# Introducing an integrated maternity care pathway for women with a history of small-for-gestational-age: Evaluation of its effect on care process and clinical outcomes

Anne C. M. Hermans[1]*, Julia Spaan[2], Marieke A.A. Hermus[3], Amber M. Hietkamp[2], Jantien Visser[2], Arie Franx[1], Jacoba van der Kooy[1]

1 Department of Obstetrics and Gynaecology, Erasmus Medical University Centre, Rotterdam, The Netherlands, 2 Department of Obstetrics and Gynaecology, Amphia Hospital, Breda, The Netherlands, 3 Midwifery Practice Oosterhout, Oosterhout, The Netherlands

* a.c.m.hermans@erasmusmc.nl

## Abstract

### Introduction

This study focusses on the implementation of an integrated care pathway for women with SGA in their obstetric history that pursues value-based healthcare. This study aims to 1) Determine whether the integrated care pathway led to a reduction in the number of antenatal secondary care consultations, as an indicator of care efficacy, and 2) compare clinical outcomes for women with a history of SGA before and after implementation of the integrated care pathway.

### Methods

Retrospective cohort study including data from pregnant women with a history of SGA within integrated maternity care organisation Annature, 2017–2020. Intervention was an integrated care pathway (2018). Pre- and post-intervention periods were compared assessing prenatal secondary care consultations, place and mode of delivery, and perinatal outcomes.

### Results

The implementation of the care pathway for pregnant women with a history of SGA led to a reduction in mean number of prenatal secondary care consultations per pregnancy from 11 in 2017–5 in 2020, and fewer inductions of labour (78 (34.2%) vs 127 (26.8%), p = 0.045). Additionally, the number of births in primary care increased (35 (15.4%) vs 136 (28.8%), p < 0.001) with no significant adverse impact on neonatal outcomes in the post-intervention period compared to the pre-intervention period.

**Data availability statement:** All relevant data are within the paper and its Supporting Information files.

**Funding:** ZonMw 10100022020013. The funders had no role in study design, data collection and analysis, decision to publish, or preparation of the manuscript.

**Competing interests:** The authors have declared that no competing interests exist.

## Conclusion

The implementation of the care pathway for pregnant women with a history of SGA resulted in a reduction in prenatal secondary care consultations and fewer inductions of labour. Additionally, the number of births in primary care increased, with no significant adverse impact on neonatal outcomes in the post-intervention period compared to the pre-intervention period.

## Introduction

Despite various initiatives aimed at improving the organization and enhancing inter professional collaboration within maternity care in the Netherlands, prompted by reports of relatively high perinatal mortality rates compared to other European countries [1], the perinatal mortality rate has remained stagnant since 2016 [2–4]. Preterm birth and low birth weight, specifically below the 10th percentile, commonly referred to as small-for-gestational-age (SGA), are the most frequent antecedents of perinatal mortality [5–8]. A key challenge in addressing this issue is organisational improvement through the implementation of a maternity care model that pursues value-based healthcare. In striving toward a patient-centred integrated maternity care system, it is crucial to strike a balance that prevents situations in which care provided 'too much, too soon' or 'too little, too late' [9]. The Netherlands employs a tiered maternity care system based on estimated risk for adverse outcomes. Primary care for low-risk pregnancies is provided by community midwives, hospital-based obstetricians manage intermediate-risk pregnancies in secondary care, and university medical centres provide tertiary care for high-risk pregnancies. Approximately 80% of pregnant women receive prenatal care across these different tiers. However, referrals from primary to secondary care can disrupt continuity, increasing the risk of losing critical medical information. Inadequate integration between these levels of care may result in duplicated procedures—such as unnecessary ultrasound examinations—thereby contributing to over-medicalisation. Therefore, The Dutch Steering Group Pregnancy and Childbirth, installed by the Ministry of Health as a response to the performance in the EURO-PERISTAT benchmark in 2008, emphasized the need for an integrated maternity care model to enhance interprofessional collaboration between primary and secondary care [1].

The integrated maternity care approach entails joint responsibility between community midwives and secondary obstetric care providers in hospital, replacing previous practices in which such intermediate-risk pregnancies were exclusively managed in secondary care. Integrated maternity care aims to provide structured, patient-centred prenatal care: Providing primary midwife-led care close to home whenever possible and secondary obstetrician-led care in hospitals when necessary.

Previous research indicates that an integrated approach to maternity care may positively influence perinatal outcomes, such as improved infant birth weight [10]. Furthermore, evidence shows that midwife-led continuity models are associated with higher maternal satisfaction while achieving perinatal outcomes comparable to other care models [11], along with potential cost-saving benefits [9,12].

In the Southwestern region, these findings eventually led to the foundation of an integrated maternity care organisation (IMCO). Within the IMCO, obstetric professionals from primary and secondary care work within an integrated patient record. [13] Furthermore, they collaboratively developed standardized protocols and pathways to enhance patient-centred care. These care pathways outline antenatal consultations and clearly set the roles of community midwives, clinical midwives, and obstetricians. Within this model, intermediate-risk pregnancies, such as those involving a history of small-for-gestational-age (SGA) infants, are managed through integrated care, with close interprofessional collaboration between primary and secondary care providers.

So far, limited studies have evaluated the effect of maternity care incorporating integrated care pathways on the pre- and perinatal care processes and clinical outcomes, particularly for women with intermediate- or high-risk pregnancies. [14,15]. This study contributes to existing literature by evaluating an integrated care pathway characterized by extensive interprofessional collaboration. The first aim of our study was to determine whether the integrated care pathway led to a reduction in the number of antenatal secondary care consultations, as an indicator of care efficacy. The secondary objective was to compare clinical outcomes for women with a history of SGA before and after the implementation of the integrated care pathway. Lastly, we compared perinatal outcomes between the pre- and post-intervention periods, stratified by whether a woman delivered an SGA neonate or a neonate with normal birth weight.

## Methods

Retrospective cohort study in which the SGA care pathway was assessed before and after its implementation. We included data from women with singleton pregnancies with a history of SGA receiving care from IMCO Annature. As of January 2018, women with a history of SGA were enrolled in the newly developed integrated SGA care pathway at the intake of a subsequent pregnancy. Prior to this, care for these women had not yet been formally structured within a standardized integrated care protocol. Consequently, the pre-intervention period was defined as women who delivered between January 2017 and September 2018, while the post-intervention period comprised women who delivered between October 2018 and December 2020. In the post-intervention group, we included women who gave birth from October 2018 until December 2020. Multiple pregnancies (n = 6) were excluded. SGA was defined as a birth weight below the 10th percentile, which was calculated using the Hoftiezer curve [16]. The birth weight percentile was automatically calculated upon registration of the birth weight, and this method remained consistent throughout the study period. The file review using patient records took place from January 8, 2022, to March 24, 2022.

### SGA care pathway: Overview

Before the implementation of the SGA care pathway, there was no structured protocol for managing intermediate-risk pregnancies, leading to considerable practice variation within the IMCO. The integrated care pathway introduced a more systematic approach by clearly assigning roles to primary and secondary care providers and scheduling multidisciplinary antenatal consultations at defined intervals.

As shown in Fig 1, the SGA care pathway includes antenatal counselling on the use of low-dose aspirin, based on the Dutch Society of Obstetrics and Gynaecology (NVOG) guideline on foetal growth restriction. It also incorporates prenatal ultrasounds at 27, 31, and 35 weeks of gestation to monitor foetal growth. Foetal growth restriction was defined as an estimated foetal weight (EFW) or abdominal circumference (AC) below the 10th percentile, or a growth deviation of ≥20 percentiles within two weeks. If no signs of foetal growth restriction were detected at the 35-week visit and no other complications were present, further management of pregnancy and labour proceeded in primary care.

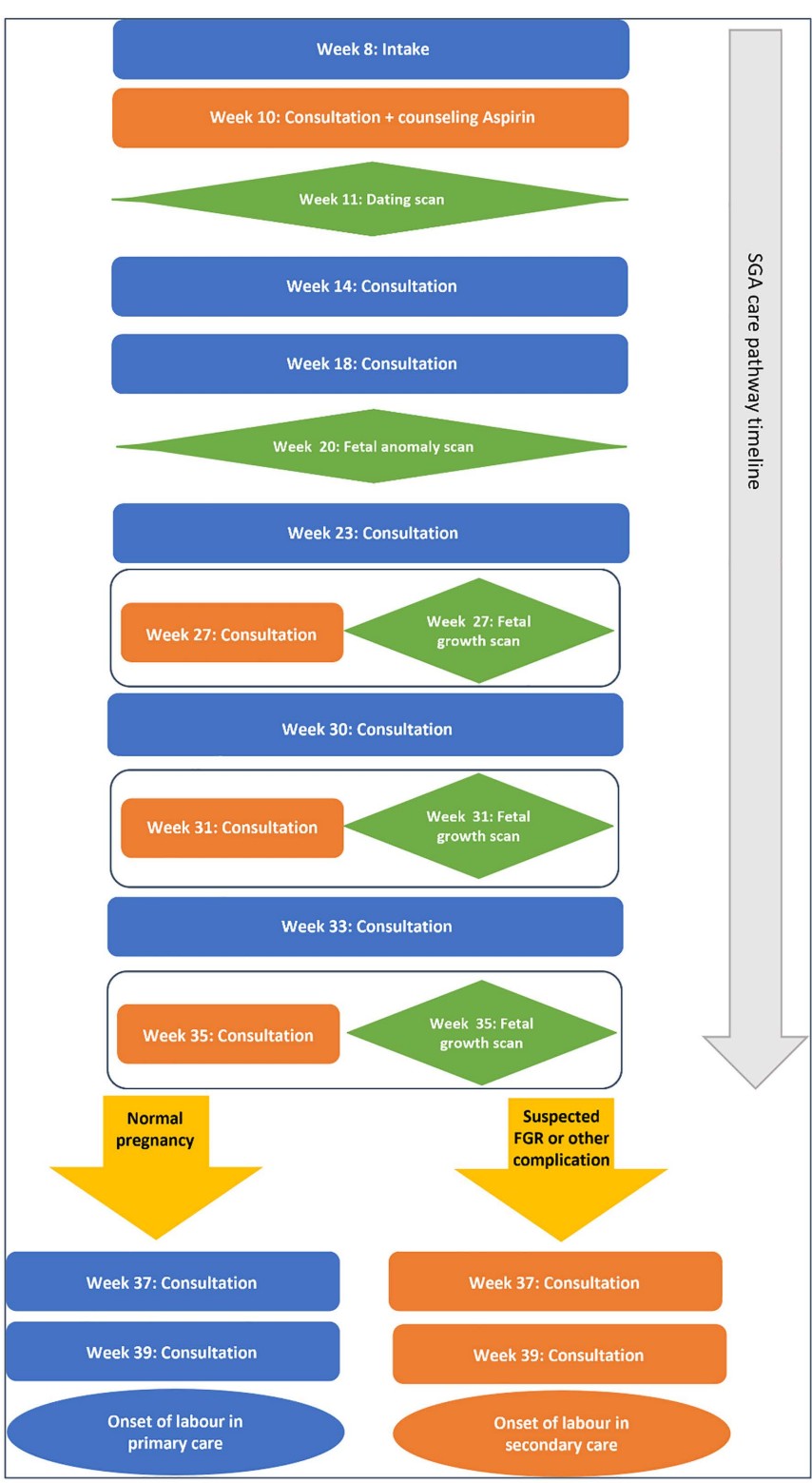

**Fig 1. Visualised SGA care pathway in which the trajectory of prenatal care is displayed in the post-intervention period.** Blue: Antenatal visit in primary care; orange: Antenatal visit in secondary care; green: prenatal ultrasound; yellow arrows: at 35 weeks of gestation the risk for foetal growth restriction (FGR) are assessed. When no foetal growth restriction or any other complication of pregnancy was detected at this antenatal visit, further follow-up of pregnancy and childbirth could take place in a primary care setting.

## Outcomes

To evaluate the impact of the SGA care pathway, two comparisons were made:

1) care and perinatal outcomes were compared between the period before (January 2017 to September 2018) and after (October 2018 to December 2020) the introduction of the integrated care pathway for women with previous SGA; and (2) perinatal outcomes were compared between the pre- and post-intervention periods, stratified by whether the neonate was SGA (the "SGA group") or had a normal birthweight (the "non-SGA group").

Baseline characteristics were compared between the pre- and post-intervention periods, and included maternal age, pre-pregnancy BMI, ethnicity, smoking at pregnancy intake, and residence in a deprived neighbourhood (defined as an area with social and infrastructural disadvantages).

Care related factors included: mean number of antenatal consultations per pregnancy per year in primary care or secondary care for the years 2017–2020. Antenatal consultations in primary care were available for analysis from 2018 onward as they were registered within the integrated patient record from 2018 onward, so the analysis focused on annual averages rather than direct pre-post comparisons.

Additional outcomes included the place and mode of delivery, based on both onset of labour and actual delivery location (primary vs. secondary care). Additionally, perinatal outcomes included the prevalence of meconium-stained amniotic fluid, an Apgar score below 7 after 5 minutes, an umbilical artery pH below 7.05, paediatric involvement, and perinatal mortality, defined as the sum of stillbirths and early neonatal deaths within the first 7 days, which is expressed per 1,000 live births. [17] Paediatric involvement was defined as any involvement of the paediatrician, which included both paediatric consultations as neonatal admissions to the ward or the NICU.

## Statistical analysis

The analyses were conducted using IBM SPSS Statistics version 25. The Chi-square test was used to compare baseline characteristics per year. For numbers <5, Fisher's exact test was applied. The threshold for statistical significance was set at alpha <0.05. All data for this study were derived from the Netherlands Perinatal Registry and from the electronic patient records of the regional hospital in the Breda region, the Netherlands. The data were anonymised prior to access for research purposes, in accordance with institutional protocols and applicable privacy regulations. The anonymisation process was carried out before the start of the analysis and was not performed by the study analysts themselves. The project leader, who is one of the co-authors, was involved in the development of the study database but did not have access to identifying information during the analysis phase.

Access to identifiable patient information was restricted to authorised healthcare professionals within the Department of Obstetrics and Gynaecology, who had such access only as part of their clinical responsibilities. The research team only received anonymised data for analysis. The retrospective use of anonymised medical records was exempt from institutional review by the Medical Ethics Committee, in compliance with Dutch national guidelines.

## Results

### Demographic characteristics

Between January 2017 and December 2020, 701 pregnant women with a history of SGA received care from IMCO Annature. Of these women, 228 received care before, and 473 received care after introduction of the SGA care pathway. Table 1 demonstrates the distribution of demographic characteristics of the 'before' and 'after' groups, with a significant decline in hypertensive disorders of pregnancy of 19 (8.3%) in the pre-intervention period compared to 15 (3.2%) in the post-intervention period (p = 0.003). No significant differences were observed in maternal age, pre-pregnancy BMI, ethnicity, living in a deprived neighbourhood or smoking.

**Table 1.** Baseline characteristics of pregnant women with small-for-gestational-age in their obstetric history. Pre-intervention period: January 2017 until September 2018; post-intervention period: October 2018 until December 2020. Data in n (%). Maternal age in years, BMI: body mass index (kg/ m²), low-dose Ascal: 80 mg per day per os.

| Demographic characteristics | Pre-intervention period (n = 228) | Post-intervention period (n = 473) | p-value |
|---|---|---|---|
| Maternal age (years) | | | 0.895 |
| ≤ 25 | 6 (2.6) | 14 (3.0) | |
| 25-34 | 150 (65.8) | 303 (64.1) | |
| ≥ 35 | 72 (31.6) | 156 (33.0) | |
| Pre-pregnancy BMI (kg/m²) | | | 0.159 |
| <25 | 130 (57.0) | 291 (61.5) | |
| 25-29.99 | 60 (26.3) | 122 (25.8) | |
| 30-34.99 | 24 (10.5) | 47 (9.9) | |
| ≥ 35 | 14 (6.1) | 13 (2.7) | |
| Ethnicity | | | 0.066 |
| Dutch | 154 (67.5) | 351 (74.2) | |
| Non-Dutch | 74 (32.50 | 122 (25.8) | |
| Deprived neighbourhood | 30 (13.2) | 48 (10.1) | 0.235 |
| Smoking | 38 (17.2) | 39 (12.6) | 0.137 |
| Hypertensive disorders of pregnancy | 19 (8.3) | 15 (3.2) | 0.003 |
| Diabetes Mellitus | 1 (0.4) | 1 (0.2) | 0.597 |
| Gestational Diabetes Mellitus | 22 (9.6) | 50 (10.6) | 0.706 |

ˣFisher's exact test. *Alpha<0.05.

## Antenatal consultations

The number of antenatal consultations is demonstrated in Table 2. The number of consultations in primary care is only available for analysis in the period 2018–2020 and remained stable over the years. The number of antenatal consultations in secondary care declined from 11 in 2017–7 in 2018–2019, and to 5 in 2020.

## Place and mode of delivery

Table 3 shows that following the introduction of the SGA care pathway, 279 women (59.0%) used low-dose aspirin, compared to 102 women (44.7%) prior to its introduction (p<0.001).

Additionally, significantly more women started labour in primary care after the pathway was introduced (214 women, 45.2%, versus 65 women, 28.5%, p<0.001). Moreover, a significantly higher number of women gave birth in primary care (136 women, 28.8%, versus 35 women, 15.4%; p<0.001). Furthermore, labour induction was performed in 127 women (26.8%) after the introduction of the SGA care pathway, compared to 78 women (34.2%) in the pre-intervention period (p=0.045). There was a trend towards lower caesarean section rates from 44 (19.3%) towards 65 (13.7%, p=0.057).

## Perinatal outcomes

Furthermore, Table 3 shows that the neonatal outcomes did not differ significantly in the pre- and post-intervention groups.

## Comparing perinatal outcomes stratified for SGA group versus non-SGA group

Table 4 presents a comparison of perinatal outcomes between the pre- and post-intervention periods, stratified by whether a woman delivered an SGA neonate (the "SGA group") or a neonate with normal birth weight (the "non-SGA group"). Table 4 shows that in the SGA-group the paediatrician was less frequently involved postpartum after the introduction of the SGA

**Table 2. Mean number of antenatal care consultations in primary care and secondary care per pregnancy for women with small-for-gestational-age in their obstetric history within integrated care organisation Annature, Breda region, the Netherlands.**

| | 2017 (n = 99) | 2018 (n = 156) | 2019 (n = 212) | 2020 (n = 222) |
|---|---|---|---|---|
| Mean number of primary care consultations per pregnancy per year | – | 8 | 8 | 8 |
| Mean number of secondary care consultations per pregnancy per year | 11 | 7 | 7 | 5 |

**Table 3. Mode of delivery and neonatal outcomes for pregnant women with small-for-gestational-age (SGA) in their obstetric history. Pre-intervention period: January 2017 until September 2018 (n = 228) versus post-intervention period: October 2018 until December 2020 (n = 473). Data are in n (%).**

| | Pre-intervention period (n = 228) | Post-intervention period (n = 473) | p-value |
|---|---|---|---|
| Obstetric characteristics | | | |
| Maternal low-dose Aspirin use | 102 (44.7) | 279 (59.0) | **<0.001*** |
| Place of birth | | | |
| Start of labour in primary care | 65 (28.5) | 214 (45.2) | **<0.001*** |
| - SGA-infant | 12/65 (18.5) | 42/214 (19.6) | 0.835 |
| Start of labour in secondary care | 163 (71.5) | 259 (54.8) | **<0.001*** |
| - SGA-infant | 56/163 (34.4) | 81/259 (31.3) | 0.510 |
| Induction of labour | 78 (34.2) | 127 (26.8) | **0.045*** |
| Place of birth in primary care | 35 (15.4) | 136 (28.8) | **<0.001*** |
| Place of birth in secondary care | 193 (84.6) | 337 (71.2) | **<0.001*** |
| Mode of birth | | | |
| Vaginal birth | 178 (78.1) | 397 (83.9) | 0.153 |
| Vacuum assisted birth[x] | 6 (2.6) | 11 (2.3) | 0.121 |
| Caesarean section | 44 (19.3) | 65 (13.7) | 0.057 |
| - Primary caesarean section | 28 (12.3) | 44 (9.5) | 0.261 |
| - Secondary caesarean section | 16 (7.0) | 21 (4.4) | 0.153 |
| Neonatal outcomes | | | |
| SGA | 68 (29.8) | 123 (26.0) | 0.287 |
| Meconium-containing amniotic fluid | 31 (13.6) | 63 (13.3) | 0.920 |
| Apgar score <7 (5 minutes post-partum)[x] | 3 (1.3) | 8 (1.7) | 1.000 |
| Ph < 7.05 (5 minutes post-partum)[x] | 2 (0.9) | 5 (1.1) | 1.000 |
| Paediatrician involvement | 123 (53.9) | 225 (47.6) | 0.114 |
| Perinatal mortality[x] | 1 (0.4) | 1 (0.2) | 0.545 |

*Alpha<0.05. [x] Fisher's exact test. =-

care pathway (pre: 63 (92.6%) versus post: 101 (82.1%), p = 0.045). In the non-SGA group, there were fewer caesarean sections after the introduction of the integrated care pathway (pre: 34 (21.3%) versus post: 46 (13.1%) p = 0.019).

## Discussion

The implementation of the care pathway for pregnant women with a history of SGA led to a reduction in prenatal secondary care consultations, and fewer labour inductions. Additionally, the number of births in primary care increased, with no significant adverse impact on neonatal outcomes in the post-intervention period compared to the pre-intervention period.

**Table 4. Mode of delivery and neonatal outcomes for pregnant women with small-for-gestational-age (SGA) in their obstetric history; SGA (n = 191) versus non-SGA (n = 510), divided into the following groups: Pre-intervention period: January 2017 until September 2018 versus post-intervention period: October 2018 until December 2020. Data are in n (%).**

| | SGA (n = 191) | | | Non-SGA (n = 510) | | |
|---|---|---|---|---|---|---|
| | Pre-intervention period (n = 68) | Post-intervention period (n = 123) | p-value | Pre-intervention period (n = 160) | Post-Intervention period (n = 350) | p-value |
| **Mode of birth** | | | | | | |
| Induction of labour | 30 (44.1) | 39 (31.7) | 0.087 | 48 (30.0) | 88 (25.1) | 0.250 |
| Vaginal birth | 56 (82.4) | 102 (82.9) | 0.827 | 122 (76.3) | 295 (84.3) | 0.065 |
| Vacuum assisted birthˣ | 2 (2.9) | 2 (1.6) | 0.617 ˣ | 4 (2.5) | 9 (2.6) | 1.000 |
| Caesarean section | 10 (14.7) | 19 (15.4) | 0.891 | 34 (21.3) | 46 (13.1) | **0.019\*** |
| - Primary caesarean sectionˣ | 5 (7.4) | 12 (10.6) | 0.608 | 23 (14.4) | 32 (9.1) | 0.077 |
| - Secondary caesarean sectionˣ | 5 (7.4) | 7 (5.7) | 0.757 | 11 (6.9) | 14 (4.0) | 0.163 |
| **Neonatal outcomes** | | | | | | |
| Meconium-containing amniotic fluid | 10 (14.7) | 15 (12.2) | 0.622 | 21 (13.1) | 48 (13.7) | 0.857 |
| Apgar score <7 (5 minutes post-partum) ˣ | 0 (0.0) | 3 (2.4) | 0.265 | 3 (1.9) | 5 (1.4) | 0.710 |
| Ph<7.05 (5 minutes post-partum) ˣ | 0 (0.0) | 2 (1.6) | 0.539 | 2 (1.3) | 3 (0.9) | 0.651 |
| Paediatrician involvement | 63 (92.6) | 101 (82.1) | **0.045\*** | 60 (37.5) | 124 (35.4) | 0.651 |
| Perinatal mortalityˣ | 0 (0.0) | 0 (0.0) | NA | 1 (0.6) | 1 (0.3) | 0.529 |

NA: not available. ˣFisher's exact test. *Alpha<0.05.

Our finding regarding the decline in mean number of antenatal secondary care consultations might be interpreted as an indicator of improved efficacy of care [18]. Furthermore, previous literature found that reduced antenatal consultations are associated with fewer interventions and improvement of perinatal outcomes in low-risk pregnancies [19]. Another randomized controlled trial in a tertiary academic centre found that a reduced frequency prenatal care model with fewer onsite appointments supplemented with virtual visit was associated with higher patient satisfaction without compromising perinatal outcomes [20].

After implementation of the care pathway, there were fewer inductions of labour (IOL) compared to the pre-intervention period. Unfortunately, the reason for IOL were not available for analysis within our study. This finding is not in line with previous research that found no association between model of maternity care and IOL rate [12].

Our study demonstrates that following the implementation of the SGA care pathway, more women gave birth in a primary care setting with similar perinatal outcomes. This shift toward primary care is significant, as it allowed more women with intermediate-risk pregnancies to deliver in primary care compared to the period before the integrated care pathway was implemented. Providing care in a primary care setting has been shown to positively impact patient experience and perinatal outcomes. [12]. Similarly, shifting care to primary maternity care settings has proven feasible for women at risk of preterm birth [20].

Regarding our secondary objectives, stratified by whether a woman delivered an SGA neonate or a neonate with normal birthweight, we found that following implementation of the SGA care pathway, the number of caesarean sections decreased in women who, in their current pregnancy, gave birth to an infant with a normal birthweight (referred to as non-SGA group). We hypothesize the integrated care pathway with a clear structure probably has facilitated this decline, although proof of causality is difficult with this 'before-after' study design. The integrated patient record enabled enhanced peer consultation between primary and secondary care providers. Together with the structured care pathway, this led to improved informational continuity of care and patient-centred care, leveraging knowledge of prior events and individual

circumstances. [21,22] This finding aligns with a systematic review showing that organisational interventions, such as team midwifery models, are associated with reduced caesarean section rates. [23]

We observed a 10% decrease in paediatric involvement in the SGA group, to 82.1% following implementation of the integrated SGA care pathway. Assessment of all 101 cases separately revealed that paediatric involvement was often omitted after uncomplicated home births with a birthweight just below the 10th percentile. In these cases, feeding advice was provided by the attending community midwife, and hospital transfer was deemed unnecessary. No adverse neonatal outcomes, such as neonatal admissions, were reported in these cases. This suggests that post-homebirth hospital transfers for glucose monitoring may pose a barrier in certain situations.

## Strengths and limitations

A key strength of this study is its provision of insights into the implementation of an integrated care pathway in clinical practice, encompassing all pregnant women in the region. From a value-based healthcare perspective, we evaluated this integrated care pathway to identify areas for future improvement. However, its retrospective design is a limitation, and we were not able to include long-term neonatal outcomes. Finally, the concurrent COVID-19 pandemic may have contributed to the observed reduction in antenatal consultations in secondary care in the end of 2020.

Future research should focus on a comprehensive cost-effectiveness analysis to provide robust evidence regarding the implementation of integrated maternity care. While this study has assessed the integrated care pathway from a value-based healthcare perspective and assessed the care process, which provide insights into care efficacy, a more comprehensive economic evaluation is required to draw more robust conclusions.

## Conclusion

The implementation of the care pathway for pregnant women with a history of SGA resulted in a reduction in antenatal secondary care consultations and fewer inductions of labour. Additionally, the number of births in primary care increased, with no significant adverse impact on neonatal outcomes in the post-intervention period compared to the pre-intervention period.

## Supporting information

**S1 File. Underlying dataset of this study.**
(XLSX)

**S2 File. STROBE checklist cohort study.**
(DOCX)

## Acknowledgments

The Netherlands Perinatal Registry, Utrecht, the Netherlands, kindly provided permission for the data analyses included.

## Author contributions

**Conceptualization:** Anne Hermans, Julia Spaan, Amber M. Hietkamp, Arie Franx, Jacoba van der Kooy.

**Data curation:** Anne Hermans, Amber M. Hietkamp, Jacoba van der Kooy.

**Formal analysis:** Anne Hermans.

**Funding acquisition:** Jantien Visser.

**Investigation:** Anne Hermans, Jacoba van der Kooy.

**Methodology:** Anne Hermans, Julia Spaan, Marieke A.A. Hermus, Amber M. Hietkamp, Jacoba van der Kooy.

**Project administration:** Anne Hermans, Jacoba van der Kooy.

**Resources:** Anne Hermans, Jacoba van der Kooy.

**Software:** Anne Hermans, Jacoba van der Kooy.

**Supervision:** Julia Spaan, Jantien Visser, Arie Franx, Jacoba van der Kooy.

**Validation:** Anne Hermans.

**Visualization:** Anne Hermans.

**Writing – original draft:** Anne Hermans.

**Writing – review & editing:** Anne Hermans, Julia Spaan, Marieke A.A. Hermus, Arie Franx, Jacoba van der Kooy.

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
