## [Decision Letter · Decision Letter 0]

25 Jun 2025

Dear Dr. Hermans,

Thank you for submitting your manuscript to PLOS ONE. After careful consideration, we feel that it has merit but does not fully meet PLOS ONE’s publication criteria as it currently stands. Therefore, we invite you to submit a revised version of the manuscript that addresses the points raised during the review process.

We look forward to receiving your revised manuscript.

Kind regards,

Ho Yeon Kim

Academic Editor

PLOS ONE

Journal Requirements:

Please confirm at this time whether or not your submission contains all raw data required to replicate the results of your study. Authors must share the “minimal data set” for their submission. PLOS defines the minimal data set to consist of the data required to replicate all study findings reported in the article, as well as related metadata and methods (https://journals.plos.org/plosone/s/data-availability#loc-minimal-data-set-definition).For example, authors should submit the following data:

Additional Editor Comments:

1. Please provide parity, the rate of hypertensive disorder in pregnancy and overt diabetes which are the main risk factors for SGA. The rate of gestational Diabetes should be also analyzed. Other risk factors for SGA should be provided in this study.

2. The indication for aspirin use in SGA needs to be stated. Aspirin use is helpful in SGA but the guideline as an indication is not established yet.

2. In order to understand the circumstance in Netherlands and be generalizable to international readers, usual antenatal practice in Dutch needs to be stated.

2. The overlap between pre and post intervention needs explanation as the reviewer recommended.

Reviewers' comments:

Reviewer's Responses to Questions

**Comments to the Author**

1. Is the manuscript technically sound, and do the data support the conclusions?

Reviewer #1: Yes

Reviewer #2: Yes

2. Has the statistical analysis been performed appropriately and rigorously?

Reviewer #1: Yes

Reviewer #2: Yes

3. Have the authors made all data underlying the findings in their manuscript fully available?

Reviewer #1: No

Reviewer #2: Yes

4. Is the manuscript presented in an intelligible fashion and written in standard English?

Reviewer #1: Yes

Reviewer #2: Yes

Reviewer #1: There is an overlap of pre-intervention and post-intervention time. As per the paper "The SGA care pathway was initially developed in December 2017, with clinical implementation occurring gradually between January and May 2018. In the pre intervention group, included women who gave birth between January 2017 and September

99 2018. In the post-intervention group, we included women who gave birth from 2018 until December

100 2020." I dont understand the reason for this overlap and why couldnt it be all together avoided?

One of the aim of the study was to see a reduction in the number of in-hospital visits? Does it mean inpatient visits?? thats quite confusing

Reviewer #2: Subject: REVIEWERS' COMMENT ON MANUSCRIPT NUMBER: PONE-D-25-11450

Title: Introducing an integrated maternity care pathway for women with a history of small-for-gestational-age: evaluation of its effect on care process and clinical outcomes

I would like to congratulate the authors on the manuscript submitted for review and publication, which focuses on an important aspect of maternal and perinatal care, focused on task shifting with optimal maternal-fetal outcome.

The article, the implementation of an integrated maternity care pathway for women with a history of small-for-gestational-age (SGA) pregnancies, evaluated the care pre and postdelivery, reducing the focus of care on the secondary health facility, except if required, to a integrated maternity care pathway, with deliveries within the primary level of care, assessing efficacy of the pathway by reduced antenatal in-hospital visits and labor inductions, and compared clinical outcomes before and after its implementation.

The outcomes were: a reduction in antenatal visits, fewer labour inductions, an increase in Primary Care births with standard monitoring of Neonatal Outcomes, reduced caesarean deliveries and improved neonatal outcomes.

The research article satisfies the following criteria:

1. The study presents the results of original research. Yes, it does.

2. Results reported have not been published elsewhere.

3. The experiments/ methods, statistics, and other analyses are performed to a high technical standard and are described in sufficient detail.

4. Conclusions are presented in an appropriate fashion and are supported by the data. Yes, this has been presented in an appropriate fashion, and it is supported by the data.

5. The article is presented in an intelligible fashion with good flow and is written in standard English.

6. The research meets all applicable standards for the ethics of experimentation and research integrity. Yes, they did not require ethical clearance, but they ensured deidentified data was used and ensured institutional protocol and national guidelines.

7. The article adheres to appropriate reporting guidelines and community standards for data availability. Yes, this was done, and a statement on how to access data was included.

However, the method section made no mention of the staff or clear documentation of the data extraction conduct or process. This can be added for completeness and easy reproducibility.

I recommend that this article be published as it meets high standards and has researched an area of importance in the management of women and their fetuses. It has provided insights into the implementation of an integrated care pathway in clinical practice, focusing on pregnant women with a history of SGA.

Thank you for the opportunity to review this manuscript for publication

**Do you want your identity to be public for this peer review?** For information about this choice, including consent withdrawal, please see our Privacy Policy

Reviewer #1: **Yes:** zaheena shamsul islam

Reviewer #2: No

---

## [Author Response · Author response to Decision Letter 1]

25 Sep 2025

Please also see the attached file ´Response to Reviewers´ in which the respond to specific reviewer and editor comments are more clearly stated:

Comments to the Author

1. Is the manuscript technically sound, and do the data support the conclusions?

Reviewer #1: Yes

Reviewer #2: Yes

2. Has the statistical analysis been performed appropriately and rigorously?

Reviewer #1: Yes

Reviewer #2: Yes

3. Have the authors made all data underlying the findings in their manuscript fully available?

Reviewer #1: No

Reviewer #2: Yes

We confirm that our submission contains all raw data required to replicate the results of our study. All relevant data, including the data required to replicate all study findings, as well as related metadata and methods), are provided within the manuscript and its Supporting Information files.

The data were pseudo anonymised prior to use, in accordance with institutional protocols and national ethical standards. The data were provided by the regional hospital and the Netherlands Perinatal Registry. One of the co-authors, who serves as the project leader, was involved in the development and preparation of the database. As the study involved retrospective analysis of pseudo anonymised medical records, it was exempt from institutional ethical review, in line with national regulations as outlined by the Central Committee on Research Involving Human Subjects (CCMO) in the Netherlands (https://english.ccmo.nl). We further confirm that the dataset is complete and that there are no restrictions on data sharing. All data relevant to the findings reported in the manuscript are available. The pseudo anonymised dataset can be made available upon reasonable request to the corresponding author, in full accordance with the journal’s data sharing policy. Should the Editor or reviewers require any further information or clarification, we would be more than happy to provide additional details as needed.

4. Is the manuscript presented in an intelligible fashion and written in standard English?

Reviewer #1: Yes

Reviewer #2: Yes

5. Review Comments to the Author

Reviewer #1: There is an overlap of pre-intervention and post-intervention time. As per the paper "The SGA care pathway was initially developed in December 2017, with clinical implementation occurring gradually between January and May 2018. In the pre intervention group, included women who gave birth between January 2017 and September

99 2018. In the post-intervention group, we included women who gave birth from 2018 until December

100 2020." I dont understand the reason for this overlap and why couldnt it be all together avoided?

As of January 2018, women with a history of SGA were enrolled in the newly developed integrated SGA care pathway at the intake of a subsequent pregnancy. Prior to this, care for these women had not yet been formally structured within a standardized integrated care protocol. Consequently, the pre-intervention period was defined as women who delivered between January 2017 and September 2018, while the post-intervention period comprised women who delivered between October 2018 and December 2020. In the post-intervention group, we included women who gave birth from October 2018 until December 2020. Consequently, there is no overlap between the two groups. We added this explanation to the method section:

Methods section, lines 97-104

As of January 2018, women with a history of SGA were enrolled in the newly developed integrated SGA care pathway at the intake of a subsequent pregnancy. Prior to this, care for these women had not yet been formally structured within a standardized integrated care protocol. Consequently, the pre-intervention period was defined as women who delivered between January 2017 and September 2018, while the post-intervention period comprised women who delivered between October 2018 and December 2020. In the post-intervention group, we included women who gave birth from October 2018 until December 2020.

One of the aim of the study was to see a reduction in the number of in-hospital visits? Does it mean inpatient visits?? thats quite confusing

We thank the reviewer for this observation and would like to clarify that we indeed assessed secondary care consultations as a measure of care efficacy.(5) By secondary care, we refer to consultations carried out by either an obstetrician or a clinical midwife in the hospital outpatient clinic.

To provide context for international readers, we have described the Dutch maternity care system and recent regional developments in care integration. The Netherlands employs a tiered maternity care system based on estimated risk for adverse outcomes: primary care for low-risk pregnancies is provided by community midwives; hospital-based obstetricians manage intermediate-risk pregnancies in secondary care; and university medical centres provide tertiary care for high-risk pregnancies. Approximately 80% of pregnant women receive prenatal care across these tiers, but referrals from primary to secondary care can disrupt continuity, risking loss of essential medical information. Poor integration between tiers may lead to redundant procedures, such as unnecessary ultrasound investigations, potentially resulting in over-medicalisation.

Our underlying hypothesis is that more integrated care, with improved cross-tier collaboration, will reduce unnecessary duplication of services and avoid non-essential consultations. Within an integrated care pathway, responsibilities and tasks of primary and secondary care providers are more clearly aligned, which facilitates efficient care allocation, prevents overlap, and promotes optimal use of healthcare resources.

Reviewer #2: Subject: REVIEWERS' COMMENT ON MANUSCRIPT NUMBER: PONE-D-25-11450

Title: Introducing an integrated maternity care pathway for women with a history of small-for-gestational-age: evaluation of its effect on care process and clinical outcomes

I would like to congratulate the authors on the manuscript submitted for review and publication, which focuses on an important aspect of maternal and perinatal care, focused on task shifting with optimal maternal-fetal outcome.

The article, the implementation of an integrated maternity care pathway for women with a history of small-for-gestational-age (SGA) pregnancies, evaluated the care pre and postdelivery, reducing the focus of care on the secondary health facility, except if required, to a integrated maternity care pathway, with deliveries within the primary level of care, assessing efficacy of the pathway by reduced antenatal in-hospital visits and labor inductions, and compared clinical outcomes before and after its implementation.

The outcomes were: a reduction in antenatal visits, fewer labour inductions, an increase in Primary Care births with standard monitoring of Neonatal Outcomes, reduced caesarean deliveries and improved neonatal outcomes.

The research article satisfies the following criteria:

1. The study presents the results of original research. Yes, it does.

2. Results reported have not been published elsewhere.

3. The experiments/ methods, statistics, and other analyses are performed to a high technical standard and are described in sufficient detail.

4. Conclusions are presented in an appropriate fashion and are supported by the data. Yes, this has been presented in an appropriate fashion, and it is supported by the data.

5. The article is presented in an intelligible fashion with good flow and is written in standard English.

6. The research meets all applicable standards for the ethics of experimentation and research integrity. Yes, they did not require ethical clearance, but they ensured deidentified data was used and ensured institutional protocol and national guidelines.

7. The article adheres to appropriate reporting guidelines and community standards for data availability. Yes, this was done, and a statement on how to access data was included.

I recommend that this article be published as it meets high standards and has researched an area of importance in the management of women and their fetuses. It has provided insights into the implementation of an integrated care pathway in clinical practice, focusing on pregnant women with a history of SGA.

Thank you for the opportunity to review this manuscript for publication

We are grateful to the reviewer for their thoughtful feedback and the considerable time and effort they invested in evaluating our manuscript.

6. PLOS authors have the option to publish the peer review history of their article (what does this mean?). If published, this will include your full peer review and any attached files.

Do you want your identity to be public for this peer review? For information about this choice, including consent withdrawal, please see our Privacy Policy.

Reviewer #1: Yes: zaheena shamsul islam

Reviewer #2: No

References

1. Leitich H, Egarter C, Husslein P, Kaider A, Schemper M. A meta-analysis of low dose aspirin for the prevention of intrauterine growth retardation. Br J Obstet Gynaecol [Internet]. 1997 [cited 2025 Aug 14];104(4):450–9. Available from: https://pubmed.ncbi.nlm.nih.gov/9141582/

2. Bujold E, Roberge S, Lacasse Y, Bureau M, Audibert F, Marcoux S, et al. Prevention of preeclampsia and intrauterine growth restriction with aspirin started in early pregnancy: A meta-analysis. Obstet Gynecol [Internet]. 2010 [cited 2025 Aug 14];116(2):402–14. Available from: https://pubmed.ncbi.nlm.nih.gov/20664402/

3. NVOG-richtlijn Foetale groeirestrictie (FGR).

4. Roberge S, Odibo AO, Bujold E. Aspirin for the Prevention of Preeclampsia and Intrauterine Growth Restriction. Clin Lab Med. 2016 Jun;36(2):319–29.

5. Peahl AF, Gourevitch RA, Luo EM, Fryer KE, Moniz MH, Dalton VK, et al. Right-Sizing Prenatal Care to Meet Patients’ Needs and Improve Maternity Care Value. Obstet Gynecol. 2020;135(5):1027–37.

---

## [Decision Letter · Decision Letter 1]

22 Dec 2025

Dear Dr. Hermans,

We look forward to receiving your revised manuscript.

Kind regards,

Ho Yeon Kim

Academic Editor

PLOS One

Journal Requirements:

Reviewer's Responses to Questions

**Comments to the Author**

Reviewer #3: All comments have been addressed

Reviewer #4: (No Response)

2. Is the manuscript technically sound, and do the data support the conclusions?

Reviewer #3: Yes

Reviewer #4: Yes

3. Has the statistical analysis been performed appropriately and rigorously?

Reviewer #3: Yes

Reviewer #4: Yes

4. Have the authors made all data underlying the findings in their manuscript fully available?

Reviewer #3: Yes

Reviewer #4: Yes

5. Is the manuscript presented in an intelligible fashion and written in standard English?

Reviewer #3: Yes

Reviewer #4: Yes

Reviewer #3: Great study - thanks for sharing with PLOS

The study highlights the importance of the implementation of an integrated care pathway.

Reviewer #4: The study is very good and recommended for the pulibication so that it will help others as well. However, there are few observations from my side attached in the manuscript.

1. Detail about the care pathway should be mentioned, any bianess during the implementation

2. Methodology section needs to be simplified little bit, it sounds more complex

**Do you want your identity to be public for this peer review?** For information about this choice, including consent withdrawal, please see our Privacy Policy

Reviewer #3: **Yes:** Martin Mueller, MD, PhD

Reviewer #4: **Yes:** Arpita Karki

---

## [Author Response · Author response to Decision Letter 2]

30 Dec 2025

Anne Hermans

Department of Gynaecology and Obstetrics

Erasmus Medical Center, Rotterdam

the Netherlands

Amsterdam, 30 December 2025

Dear Editor,

Attached, please find our manuscript entitled “Introducing an integrated maternity care pathway for women with a history of small-for-gestational-age: evaluation of its effect on care process and clinical outcomes” which we would like to be considered for publication in PLOS One medical journal.

We would like to thank the reviewers for their time and valuable comments. We have carefully considered all points raised and have revised and improved the manuscript accordingly.

Thank you for considering our paper for publication in your journal.

We believe that our study results may be of interest to a wide audience representing PLOS One medical journal readers that include clinicians involved in maternity care. Please do not hesitate to reach out if you have any questions.

Yours sincerely (on behalf of all authors),

Anne Hermans, MD

Journal Requirements:

Reviewer's Responses to Questions

Comments to the Author

1. If the authors have adequately addressed your comments raised in a previous round of review and you feel that this manuscript is now acceptable for publication, you may indicate that here to bypass the “Comments to the Author” section, enter your conflict of interest statement in the “Confidential to Editor” section, and submit your "Accept" recommendation.

Reviewer #3: All comments have been addressed

Reviewer #4: (No Response)

2. Is the manuscript technically sound, and do the data support the conclusions?

Reviewer #3: Yes

Reviewer #4: Yes

3. Has the statistical analysis been performed appropriately and rigorously?

Reviewer #3: Yes

Reviewer #4: Yes

4. Have the authors made all data underlying the findings in their manuscript fully available?

Reviewer #3: Yes

Reviewer #4: Yes

5. Is the manuscript presented in an intelligible fashion and written in standard English?

Reviewer #3: Yes

Reviewer #4: Yes

6. Review Comments to the Author

Reviewer #3: Great study - thanks for sharing with PLOS

The study highlights the importance of the implementation of an integrated care pathway.

Reviewer #4: The study is very good and recommended for the pulibication so that it will help others as well. However, there are few observations from my side attached in the manuscript.

1. Detail about the care pathway should be mentioned, any bianess during the implementation

We thank the reviewer for this valuable comment. From the moment of implementation, all women with a history of delivering an SGA neonate were systematically enrolled in the SGA care pathway at the intake of a subsequent pregnancy. As of January 2018, these women were included in the newly developed integrated SGA care pathway. Prior to implementation, care for women with a previous SGA pregnancy was not formally structured within a standardized integrated care protocol, which resulted in variation in clinical management.

Accordingly, the pre‑intervention period was defined as women who delivered between January 2017 and September 2018, whereas the post‑intervention period included women who delivered between October 2018 and December 2020. This approach allowed for a clear distinction between routine care prior to pathway implementation and care provided within the integrated SGA care pathway.

To reduce the risk of implementation bias, all eligible women were enrolled based on obstetric history, without selective inclusion. The only exclusion applied to multiple pregnancies, as the care pathway was specifically designed for singleton pregnancies. Figure 1 illustrates the care timeline in detail, including the scheduled antenatal consultations and the distinction between primary and secondary care settings.

2. Methodology section needs to be simplified little bit, it sounds more complex

We thank the reviewer for this remark as well, and have revised the method section as follows (lines 110-180)

SGA care pathway: overview

Before the implementation of the SGA care pathway, there was no structured protocol for managing intermediate-risk pregnancies, leading to considerable practice variation within the IMCO. The integrated care pathway introduced a more systematic approach by clearly assigning roles to primary and secondary care providers and scheduling multidisciplinary antenatal consultations at defined intervals.

As shown in Fig 1, the SGA care pathway includes antenatal counselling on the use of low-dose aspirin, based on the Dutch Society of Obstetrics and Gynaecology (NVOG) guideline on foetal growth restriction. It also incorporates prenatal ultrasounds at 27, 31, and 35 weeks of gestation to monitor foetal growth. Foetal growth restriction was defined as an estimated foetal weight (EFW) or abdominal circumference (AC) below the 10th percentile, or a growth deviation of ≥20 percentiles within two weeks. If no signs of foetal growth restriction were detected at the 35-week visit and no other complications were present, further management of pregnancy and labour proceeded in primary care.

Outcomes

To evaluate the impact of the SGA care pathway, two comparisons were made:

1) care and perinatal outcomes were compared between the period before (January 2017 to September 2018) and after (October 2018 to December 2020) the introduction of the integrated care pathway for women with previous SGA; and (2) perinatal outcomes were compared between the pre- and post-intervention periods, stratified by whether the neonate was SGA (the “SGA group”) or had a normal birthweight (the “non-SGA group”).

Baseline characteristics were compared between the pre- and post-intervention periods, and included maternal age, pre-pregnancy BMI, ethnicity, smoking at pregnancy intake, and residence in a deprived neighbourhood (defined as an area with social and infrastructural disadvantages).

Care related factors included: mean number of antenatal consultations per pregnancy per year in primary care or secondary care for the years 2017 to 2020. Antenatal consultations in primary care were available for analysis from 2018 onward as they were registered within the integrated patient record from 2018 onward, so the analysis focused on annual averages rather than direct pre-post comparisons.

Additional outcomes included the place and mode of delivery, based on both onset of labour and actual delivery location (primary vs. secondary care). Additionally, perinatal outcomes included the prevalence of meconium-stained amniotic fluid, an Apgar score below 7 after 5 minutes, an umbilical artery pH below 7.05, paediatric involvement, and perinatal mortality, defined as the sum of stillbirths and early neonatal deaths within the first 7 days, which is expressed per 1,000 live births.(17) Paediatric involvement was defined as any involvement of the paediatrician, which included both paediatric consultations as neonatal admissions to the ward or the NICU.

7. PLOS authors have the option to publish the peer review history of their article (what does this mean?). If published, this will include your full peer review and any attached files.

Do you want your identity to be public for this peer review? For information about this choice, including consent withdrawal, please see our Privacy Policy.

Reviewer #3: Yes: Martin Mueller, MD, PhD

Reviewer #4: Yes: Arpita Karki

---

## [Editor Report · Decision Letter 2]

2 Jan 2026

Introducing an integrated maternity care pathway for women with a history of small-for-gestational-age: evaluation of its effect on care process and clinical outcomes

PONE-D-25-11450R2

Dear Dr. Hermans,

We’re pleased to inform you that your manuscript has been judged scientifically suitable for publication and will be formally accepted for publication once it meets all outstanding technical requirements.

Kind regards,

Ho Yeon Kim

Academic Editor

PLOS One
---

## [Editor Report · Acceptance letter]

PONE-D-25-11450R2

PLOS One

Dear Dr. Hermans,

I'm pleased to inform you that your manuscript has been deemed suitable for publication in PLOS One. Congratulations! Your manuscript is now being handed over to our production team.

Kind regards,

on behalf of

Professor Ho Yeon Kim

Academic Editor

PLOS One